# Innovative Approach to Enhance Bioavailability of Birch Bark Extracts: Novel Method of Oleogel Development Contrasted with Other Dispersed Systems

**DOI:** 10.3390/plants13010145

**Published:** 2024-01-04

**Authors:** Laura Andze, Sanita Vitolina, Rudolfs Berzins, Janis Rizikovs, Daniela Godina, Arturs Teresko, Solveiga Grinberga, Eduards Sevostjanovs, Helena Cirule, Edgars Liepinsh, Aigars Paze

**Affiliations:** 1Latvian State Institute of Wood Chemistry, 27 Dzerbenes Street, LV-1006 Riga, Latvia; sanita.vitolina@kki.lv (S.V.); rudis.berzins@gmail.com (R.B.); janis.rizikovs@kki.lv (J.R.); daniela.godina@kki.lv (D.G.); aigars.paze@kki.lv (A.P.); 2ZS DOKTUS, 22 Pavila Street, LV-4101 Cesis, Latvia; arturs_teresko@inbox.lv; 3Latvian Institute of Organic Synthesis, Aizkraukles Street 21, LV-1006 Riga, Latvia; solveiga@osi.lv (S.G.); eduards@osi.lv (E.S.); helena.cirule@farm.osi.lv (H.C.); ledgars@farm.osi.lv (E.L.)

**Keywords:** betulin, lupeol, betulinic acid, birch bark extracts, triterpenes, hydrogel, oleogel, bioavailability

## Abstract

Birch outer bark extract (BBE), containing pentacyclic triterpenes such as betulin, lupeol, and betulinic acid, is a widely recognized natural product renowned for its diverse pharmacological effects. However, its limited water solubility restricts its bioavailability. Therefore, the main objective is to enhance the bioavailability of BBE for pharmaceutical use. In this study, we aimed to develop a dispersion system utilizing a unique oleogel-producing method through the recrystallization of BBE from an ethanol solution in the oil phase. We generated an oleogel that demonstrates a notable 42–80-fold improvement in betulin and lupeol peroral bioavailability from BBE in Wistar rats, respectively. A physical paste-like BBE hydrogel developed with antisolvent precipitation showed a 16–56-fold increase in the bioavailability of betulin and lupeol from BBE in rat blood plasma, respectively. We also observed that the repeated administration of the BBE oleogel did not exhibit any toxicity at the tested dose (38.5 mg/kg betulin, 5.2 mg/kg lupeol, 1.5 mg/kg betulinic acid daily for 7 days). Betulin and betulinic acid were not detected in rat heart, liver, kidney, or brain tissues after the peroral administration of the oleogel daily for 7 days. Lupeol was found in rat heart, liver, and kidney tissues.

## 1. Introduction

Herbal extracts are widely used in the formulation of various health products, including dietary supplements and drugs [1]. Birch outer bark extracts (BBEs) have numerous pharmacological properties [2] and exhibit anti-inflammatory, antioxidant, and anticancer properties [3,4], as well as being effective in therapy for chronic hepatitis C [5].

BBE extracts contain several active components, among which betulin, lupeol, and betulinic acid are the most abundant and well-studied triterpenes [6,7]. Betulin has been reported to be effective against inflammation and oxidative stress [8,9,10]. Additionally, it has demonstrated antidiabetic [11], hepatoprotective [12], cardioprotective [13], and neuroprotective effects [14]. Lupeol may help to manage osteoporosis and possesses potential cardioprotective and skin-protective properties [15]. Betulinic acid exhibits diverse biological activities, including antitumor and anti-inflammatory effects [2,9,16]. Despite the remarkable medicinal properties of betulin, lupeol, and betulinic acid, their therapeutic efficacy is limited by their low bioavailability. Therefore, there is a need to improve the bioavailability of BBE for improved clinical efficacy.

Various factors affect the bioavailability of BBEs, including solubility, stability, and permeability [2]. Another factor that influences the bioavailability of BBEs is permeability or absorption. The absorption and systemic uptake of betulin, lupeol, and betulinic acid depend on their physicochemical properties, such as size, shape, charge, and lipophilicity. BBE contains lipophilic compounds that limit their bioavailability since they are not easily absorbed in the gastrointestinal tract [17,18]. Moreover, BBE components are susceptible to degradation and oxidation, which further reduce their bioavailability [7,19]. To address the low bioavailability of BBEs, various approaches have been developed, including physical modification, chemical modification, and formulation strategies. Physical modification techniques, such as micronization [20], nanoemulsification [21,22,23], and solid dispersion [20,24], are used to increase the surface area and reduce the particle size of BBE. Chemical modifications, for example, esterification [25,26,27], glycosylation, acylation [10,17,28], and the synthesis of prodrugs, aim to improve the chemical stability and lipophilicity of BBEs [29]. Formulation strategies, including encapsulation, complexation, and co-administration with absorption enhancers, aim to improve the delivery and absorption of BBEs [10,17,30].

Hydrogels can release drugs in a controlled manner, providing sustainable and targeted delivery and bioavailability by solubilizing the extracts and enhancing their absorption in the gastrointestinal tract. Hydrogels can also act as a barrier, preventing the extracts from coming into contact with oxygen and other reactive species that can cause their degradation [31,32,33]. 

Chemical hydrogels consist of hydrophilic components that are covalently crosslinked to form a three-dimensional polymeric network. Studies on chemical hydrogels of BBEs have been conducted using natural polymers, specifically chitosan and sodium alginate, with the aim of enhancing their properties [34,35]. On the other hand, physical or so-called reversible hydrogels rely on dynamic and reversible non-covalent crosslinking, like hydrogen bonding and ionic or electrostatic interactions [36]. Physical hydrogels are homogeneous, but they lack stability when compared to chemical hydrogels. The intermolecular interactions in physical gels are relatively weak, making them reversible. Accordingly, physical hydrogels possess lower mechanical strength. Because of their reversible nature, these hydrogels are highly attractive for the inclusion of bioactive substances [37]. In our earlier investigations, physical hydrogels with a paste-like consistency were produced using the antisolvent precipitation technique for BBE [38,39]. This process resulted in a reduction in BBE particle sizes, potentially enhancing bioavailability. Importantly, no toxic effects on skin cells were observed [40].

A promising approach is to use an oleogel as a drug formulation to enhance the bioavailability of lipophilic compounds. Several factors might be involved, such as the slow release of active compounds and the formation of smaller particles with increased surface area. For instance, oleogels can entrain the active compounds in their oil phase and release them slowly over an extended period, which can improve their absorption in the body [29,41,42]. Research has shown that BBEs can form oleogels, thus increasing the possibilities of using BBEs in medical applications [43,44]. Studies on BBE oleogels, both with [42] and without gelling agents [43,45], indicate that the oil phase provides better bioavailability of BBE [46], mainly because the solubility of BBE in the oil phase is up to 17,000 times better than in water [47].

An efficient and sensitive method for the quantification of betulin, betulinic acid, and lupeol is essential for pharmacokinetic studies. Liquid chromatography–electrospray tandem mass spectrometry (LC–ESI/MS/MS) of intact molecules, offering high selectivity and sensitivity, can only be applied to acidic triterpenes such as betulinic acid [48]. However, neutral triterpenes possessing only a hydroxyl group (such as betulin and lupeol) are not efficiently ionized by ESI. Precolumn derivatization, widely used in HPLC, has been used to enhance the detection sensitivity and ionization efficiency of compounds for MS detection. Betulin and lupeol have been successfully ionized and fragmented after derivatization with p-toluenesulfonyl isocyanate (PTSI) and quantified using LC–ESI/MS/MS [49,50].

The aim of this study is to assess the bioavailability of a birch *(Betula pendula)* outer bark extract oleogel prepared by a novel method of BEE crystallization from an ethanol solution in the oil phase [51] and compared to the BBE bioavailability of a physical paste-like hydrogel developed with antisolvent precipitation [39] and BBE suspensions in water and oil. The study evaluated and compared the exposures of lupeol, betulin, and betulinic acid administered simultaneously. As a model animal, Wistar rats were used. The bioavailability of betulin, lupeol, and betulinic acid in rat blood plasma was determined simultaneously using an adapted LC/MS/MS method after peroral administration of the BBE oleogel.

## 2. Results and Discussion

### 2.1. Visual Evaluation of BBE Hydrogel and Oleogel

The first image shows a visual representation of the hydrogel (left) and oleogel (right) containing 4% BBE in water and 8% BBE in sunflower oil, respectively (Figure 1).

### 2.2. Particle Size Distribution of BBE Dispersion Systems

BBE particle sizes were determined using two different dynamic light scattering (DLS) methods, a Zetasizer Nano-ZS instrument and a Mastersizer 3000 laser diffraction system, suitable for the water and oil phases, respectively. Table 1 summarizes the mean size of BBE particles in the aqueous phase, showing a narrow distribution of results. 

The BBE particles in the oil fraction display a tri-modal size distribution, characterized by the presence of three distinct peaks. The results in Table 1 reveal a difference in the particle sizes of the hydrogel and BBE powder in the water suspension, with the hydrogel exhibiting four times smaller particles. Additionally, the particle size of BBE powder in the oil suspension is six times larger than in the oleogel. The overall preparation of the hydrogel and oleogel formulations was successful, forming stable systems with smaller particles than in suspensions, and they could be further used for bioavailability testing.

### 2.3. Oxidative Stability of BBE Oleogel 

Due to their high oil content, oleogels are characterized by instability in an oxygen environment, so at the beginning of the study, an oxidative stability test of the oleogel was performed, which was compared with a pure sunflower oil sample and an oil sample containing TBHQ (tertiary butylhydroquinone). 

The obtained results (Figure 2 and Table 2) show that the oleogel is oxidatively stable, and the polyphenols in BBE provide antioxidant properties with a protection factor of 3.2, equal to or even better than TBHQ, which is often used in food.

### 2.4. LC/MS/MS Method for Simultaneous Determination of BBE 

The developed bioanalytical method was suitable for the quantitative determination of betulin, betulinic acid, and lupeol in rat blood plasma. The procedure was selective enough to quantify 5 ng/mL of betulin and betulinic acid and 10 ng/mL of lupeol in rat plasma. The linear detection range was from 5 ng/mL to 500 ng/mL for all analytes. Figure 3 shows the characteristic chromatograms.

### 2.5. In Vivo Analysis of BBE

#### 2.5.1. Concentrations of BBE Compounds in Rat Plasma

The measured concentrations of betulin in plasma samples are depicted in Figure 4. The peak betulin concentrations in all tested groups occurred at the 2 h time point. The highest concentrations of betulin in the plasma were observed following oral oleogel administration. In this group, the betulin plasma concentration reached 344 ng/mL, surpassing the betulin concentrations in the hydrogel group by at least 3 times. When compared to the BBE suspension in oil, oleogel administration resulted in a 6 times higher betulin concentration. This difference is attributed to the fact that the oleogel contains smaller BBE particles than the oil suspension, resulting in a larger surface area and, consequently, improved bioavailability [11,23,38,39,40,46]. 

The lowest peak concentration of betulin was observed in rat plasma following the administration of the BBE suspension in water, a concentration that was as much as 5-fold lower than in the oil suspension group. The disparity in the betulin concentration between the oleogel and the hydrogel, as well as between the BBE suspensions in oil and water, indicates that an oil-containing system enhances betulin uptake. The solubility of BBE in water is very low (0.25 µg/mL), while in sunflower oil, it is 17,000 times higher (4.4 mg/mL) [47]. This stark contrast accounts for the improved bioavailability in the oil phase. Betulin concentrations in the oleogel half-dose group reached a peak concentration approximately half that of the full-dose oleogel (Figure 4). Thus, we demonstrate that the bioavailability of botulin from the oleogel is dose-dependent within the tested dose range. On the last day of the 7-day daily administration, the peak betulin concentration was lower, and the pharmacokinetic (PK) profile exhibited substantial differences.

A previous study by Jager et al. on the pharmacokinetics of betulin after the administration of a BBE suspension in sesame oil showed a dose-independent maximum betulin level of 130 ng/mL 4 h after administration [52]. In our study, after the administration of BBE in the oleogel, betulin reached a 3 times higher concentration in blood plasma than in the Jager et al. study. However, after the administration of the BBE oil suspension, a 2.5 times lower betulin concentration was observed at the same time point than in Jager et al.’s study. It should be noted that the studies used different strains of rats, which may produce different results [53]. In Pozharitskaya’s investigation, the use of betulin nanoparticles at a dose 3 times lower (25.2 mg/kg) demonstrated significantly higher bioavailability results (15.5 µg/mL) in blood plasma compared to the data observed in our study using the oleogel. It should be noted that in Pozharitskaya’s research, betulin was administered through the endotracheal route, which notably enhances the bioavailability indicators, and the studies are not comparable. This finding is further supported by Pozharitskaya’s data, where directly injected betulin powder reached 6.9 µg/mL in blood plasma, which is almost 1000 times higher than in the case of the aqueous suspension of BBE powder in our study. Additionally, it is important to mention that Pozharitskaya’s study employed a distinct rat species and administered pure betulin instead of a mixture of extracts. In Pozharitskaya’s study, betulin nanoparticles increased the bioavailability by 2.5 times compared to betulin powder, but in our study, the oleogel system increased the bioavailability by 42 times compared to the powder suspension in water [54]. In Zhao et al.’s study, they orally administered betulin nanoparticles obtained by the antisolvent precipitation method, and the peak plasma concentration of betulin was 6 µg/mL, which indicates a 20 times greater bioavailability than in our study. It should be noted that in the study by Zhao et al., pure betulin nanoparticles were administered instead of a BBE mixture [11].

It is possible that, due to their structural similarity (Figure 5), betulin, lupeol, and betulinic acid may compete among themselves for the absorption processes, similar to the case for the solubility balance [43].

When comparing the total betulin exposure based on the AUC (area under the curve) across various BBE-containing dispersed systems, the highest betulin exposure was observed in the oleogel group rats, exceeding hydrogel exposure by 2.6-fold (Figure 6). This discrepancy is attributed to the superior solubility of BBE in oil compared to water [43,47]. Furthermore, betulin exposure in the oleogel group was 5-fold higher than in the oil suspension, while its exposure in the hydrogel group was 16-fold higher than in the water suspension group. This can be explained by the smaller particle sizes in oleogel and hydrogel dispersion systems [40]. The oleogel at a half dose reached 2.8-fold lower betulin exposure than the full-dose oleogel after a single administration (Figure 6). 

The measured concentrations of lupeol in plasma samples are shown in Figure 7. The peak lupeol concentrations in all tested groups, except the water suspension, were at a 4 h time point. The highest plasma concentrations of lupeol were measured after oleogel administration. In this group, the lupeol plasma concentration reached 610 mg/mL, which exceeded the lupeol concentrations in all other groups but not by the same margin as betulin. Thus, the peak lupeol plasma concentration in the hydrogel group was only 20% lower than in the oleogel group. However, in comparison to the BBE suspension in oil, oleogel administration induced an 8 times higher lupeol concentration. Similar to betulin, the lowest peak concentration of lupeol was found in rat plasma after the administration of the BBE suspension in water, which was as much as 50-fold lower than in the oleogel group.

Peak lupeol concentrations in both measurements after the first and last administrations in the oleogel half-dose group were comparable, and approximately half of the lupeol plasma concentration was achieved after the administration of the oleogel at a full dose (Figure 7). This implies that the bioavailability of lupeol is dose-dependent within the tested dose range. Nevertheless, on the last day of seven administrations, the peak lupeol concentration was slightly lower, and the PK profile exhibited differences. Similar to the betulin PK profile, the lupeol profile also indicates a slower but longer uptake of BBE compounds (Figure 5 and Figure 7).

Similar results were achieved in a study by Priyanka et al., where a 5 times higher dose of orally administered lupeol solid lipid nanoparticles reached 696 ng/mL of lupeol in the rat blood plasma [46]. The results of our study demonstrate that the oleogel achieves a 4 times higher bioavailability of lupeol than what was observed in Priyanka et al.’s research. In the investigation conducted by Cháirez-Ramírez et al. [45], lupeol was orally administered in olive oil at a concentration that was 20 times higher (200 mg/kg) than in our study. Nevertheless, our findings revealed only a 10-fold lower lupeol concentration in blood plasma (600 ng/mL) compared to the Cháirez-Ramírez study (6 ng/µL) [45]. Hence, the oleogel formulation resulted in a two-fold increase in the bioavailability of lupeol, as compared to the findings of Cháirez-Ramírez et al.’s study [45]. It is essential to note that the experimental protocol of Cháirez-Ramírez et al.’s research involved orally administering pure lupeol to mice, which may hinder direct comparisons with our study using rats [45]. Considering the similarities between lupeol and betulin, it is possible that they may influence bioavailability, similar to the solubility balance. In general, lupeol demonstrates significantly better bioavailability than betulin, partly due to its superior solubility in both water and oil compared to betulin and betulinic acid [43,45,47,52].

The assessment of total lupeol exposure (AUC) in different dispersed BBE systems exhibited superior lupeol exposure in the oleogel group rats, registering 40% lower bioavailability results than in the study conducted by Priyanka et al [46]. However, it should be noted that the Priyanka et al. study employed a 5 times higher dose of lupeol than was used in our study [46]. This indicates that the oleogel can be considered the most effective formulation in terms of lupeol bioavailability. These data are summarized in Figure 8. Oleogel exposure exceeded hydrogel exposure by 42%. Lupeol exposure in the oleogel group was 5- and 80-fold higher than exposure in the oil suspension and water suspension groups, respectively. The oleogel at a half dose after a single administration reached exactly 2-fold lower lupeol exposure than in the oleogel full-dose group. Exposure after seven administrations was 1.5-fold higher than after a single administration of half a dose.

Lupeol exposure was about 3 times higher than betulin exposure. Considering the lower dose of lupeol, its bioavailability from the oleogel is 20 times better than betulin bioavailability. Betulinic acid content in rat blood plasma was not detected or was below the limit of quantification for all BBE-containing dispersion systems at all time points. This indicates the low bioavailability of betulinic acid, but it should be noted that the administered dose of betulinic acid was 3 times lower than that of lupeol and 26 times lower than that of betulin. Taking into account that the dose of lupeol is only 3 times higher than that of betulinic acid, but the detection limit is 5 ng/mL for betulinic acid and 10 ng/mL for lupeol, it can be concluded that betulinic acid has the lowest bioavailability if the substances (betulin, lupeol, and betulinic acid) are administered at the same time; otherwise, even at such a low dose, we could assess betulinic acid in blood plasma. Veber et al.’s research supports that a decrease in polar surface area exhibits a positive correlation with an increase in the permeation rate, instead of lipophilicity, and that lower molecular weight is associated with higher oral bioavailability [55].

Additionally, their study demonstrates that oral bioavailability is further linked to lower rotatable bond counts, lower hydrogen bond counts, and lower polar surface area. Based on this information, it is reasonable to conclude that the polar surface area of a compound plays a crucial role in determining its bioavailability [55]. For instance, the polar surface areas of lupeol, betulin, and betulinic acid are approximately 20, 40, and 60 A^2^, respectively, and their molecular weights are 426, 442, and 457, respectively. This likely influenced the bioavailability of lupeol in our tests and explains the low bioavailability of betulinic acid.

#### 2.5.2. Content of BBE in the Tissues

After repeated (half dose daily for 7 days) administrations, tissues were collected for the determination of BBE compounds. 

Betulin and betulinic acid (level of quantification 50 ng/g) were not found in any of the tested tissues 24 h after the last administration. The results of the lupeol tissue content are presented in Figure 9. 

A substantial amount of lupeol was detected in the heart and kidneys. In both tissues, a similar amount (about 250 ng/g tissue) of lupeol was detected. In comparison, in the liver, only about 21 ng/g tissue was detected. Lupeol was not found in the brain, indicating that it was not able to cross the blood–brain barrier.

#### 2.5.3. Evaluation of Oleogel Safety

No adverse reactions were observed during the administration of the oleogel and other preparations. Following seven administrations of the oleogel at a half dose, the rats underwent examination for macroscopic changes in organs and measurements of clinical chemistry markers indicative of kidney and liver functionality. No macroscopic changes were observed in the liver, kidney, or other organs. Similarly, clinical chemistry markers showed no significant increases in the treatment group (Table 3). The oleogel demonstrated safety for the liver and kidney at the tested dose. Similar results regarding safety and non-toxicity have been reported in studies by other authors in which no toxic reaction was detected at the tested doses [7,42,56].

## 3. Materials and Methods

### 3.1. Materials

Dry birch *(Betula pendula)* outer bark (A/S Latvijas Finieris, Latvia) was used as the raw material for the extraction process with >95 vol% ethanol to obtain the hydrogel, oleogel, and dry powder of BBE. Subsequently, water and oil suspensions were obtained from the powder of BBE. Applied reference standards, solvents, and materials are summarized in Table 4.

### 3.2. Preparation of BBE Powder

Using 3 kg of dried birch *(Betula pendula)* outer bark with a dry matter content of 95–96% and a particle size of 0.5–3.15 mm, an extraction process was conducted in a 30 L originally constructed reactor using 18 L of ethanol. The reactor is equipped with a steam heating jacket and a reflux condenser. The extraction process involved boiling the mixture for 1 h at the boiling temperature of ethanol (78.3 °C), with independent stirring at 80 rpm. After boiling, the resulting solution was filtered. An additional 5 L of ethanol was used to wash the filtrate through a 100 µm filter bag. 

This process resulted in approximately 14 L of filtrate. The obtained filtrate solution was then used for oleogel production. The filtrate was evaporated in a 20 L round-bottom flask on a Hei-VAP industrial vacuum rotary evaporator (Laborota 4003, Heidolph, Germany) to obtain a wet-paste-like extract. To obtain extract powder, the paste-like substance was dried in an oven at 80°C temperature until a constant mass was achieved. Subsequently, the dried substance was ground in a ball mill and fractionated to pass through 125 µm sieves. A water-insoluble BBE powder was obtained with a bulk density of 497.6 = 14.4 g/L and a solubility in alcohol of 55.5 = 3.7 g/L. The main chemical characteristics are summarized in Table 5.

### 3.3. Preparation of BBE Powder Suspension in Water and Oil 

In order to obtain a suspension of BBE in water, 8 g of BBE powder (obtained by the method described above) was mixed with 100 mL of water containing swollen 1% hydroxymethylpropylcellulose for 2 min using a T10 basic Ultra-Turrax homogenizer (IKA) at 7000 rpm. To prepare the oil suspension, 100 mL of sunflower oil and 8 g of BBE powder (obtained by the method described above) were mixed for 2 min using a T10 basic Ultra-Turrax homogenizer (IKA) at 7000 rpm.

### 3.4. Obtainment of BBE Hydrogel

The hydrogel was prepared using a previously established antisolvent precipitation method [39]. In a 1 L glass reactor equipped with an electric stirrer, heater, and reflux condenser, 55 g of BBE powder (determined as maximum saturation in a previous study [39]) was dissolved in 1 L of boiling ethanol. Then, 250 mL of the resulting hot solution was slowly poured into 1750 mL of distilled water while continuously stirring. The finely dispersed particles formed within a 12.5% *v*/*v* ethanol solution were concentrated using a Buchner funnel and filter paper (Whatman, grade 3, medium flow) with a particle retention size of 6 µm under vacuum. Once the process was complete, the homogeneous hydrogel was carefully separated from the filter paper and stored at 3 °C in a refrigerator. The resulting hydrogel was homogeneous and devoid of any particulate matter. The dry matter content of the hydrogel sample was measured to be approximately 5%.

### 3.5. Preparation of BBE Oleogels

A total of 178 g of a BBE solution containing approximately 4.5% ethanol was added to 92 g of sunflower oil in a 2 L round flask. The round flask was then connected to a vacuum rotary evaporator (Laborota 4003, Heidolph, Germany) to separate the ethanol from the oil. The ethanol was condensed and collected in a pouring flask, with a water bath temperature of 50 °C, vacuum of 100 mbar, and flask rotation speed of 80 rpm. The distillation rate averaged between 10 and 15 mL/min. To ensure maximum separation of ethanol from the oil, the distillation parameters were increased after distilling about 150 g of ethanol. The water bath temperature was raised to 95 °C, the vacuum was reduced to 20 mbar, and the flask rotation speed remained at 80 rpm. Consequently, a mixture of BBE colloidal particles was obtained and dispersed in sunflower oil. This mixture was further homogenized using a Pro Mix Titanium homogenizer (Philips, Amsterdam, The Netherlands) for 30 s at an average speed of 6000 rpm. The purpose of this step was to create a uniform oleogel. The resulting oleogel had an average dry matter content of 8% and ethanol content of 0.2 wt%. It was then transferred to a storage container and allowed to thicken upon cooling.

### 3.6. Particle Size Distribution

To determine average particle sizes, dynamic light scattering (DLS) measurements were employed. 

Water dispersion systems of BBE, like the hydrogel and BBE powder suspension in water, were measured using a Zetasizer Nano-ZS instrument from Malvern Instruments, Malvern, UK. To prepare the sample for analysis, a weight of BBE powder or hydrogel containing 10 mg of oven-dried BBE was dispersed in 100 mL of water using a T10 basic Ultra-Turrax homogenizer (IKA) for 1 min at 11,500 rpm. After homogenization, ultrasonication with a Hielscher UP200Ht (Heilscher Ultrasonics, Teltow, Germany) at 20% amplitude was employed for an additional minute. All measurements were taken immediately following homogenization at a temperature of 25 °C in triplicate.

To measure the particle size distribution of the oleogel and oil suspension, the Mastersizer 3000 laser diffraction system with a Hydro-EV mixer from Malvern Instruments, Malvern, UK, was utilized. The system can measure particles with a range of 10 nm–3.5 mm. In this particular study, a previously prepared oleogel of sunflower oil and BBE at a concentration of 8% were used. To ensure accurate measurement, the samples were diluted in a ratio of 1:10 in the oil phase. The sample was then stirred in sunflower oil to ensure homogeneous dispersion before being added to a cuvette for measurement. Optical concentration was set at 25–30%. The oleogels used for particle sizing were stored at 25 °C.

### 3.7. Oxidative Stability of BBE Oleogel

To determine the oxidative stability, the oleogel was compared with a sunflower oil alone (blank sample) and a sample of sunflower oil with the addition of TBHQ (tertiary butylhydroquinone), which is widely used as an effective food antioxidant. TBHQ was added to the oil at a concentration equivalent to the concentration of polyphenols in the oleogel—0.33% (determined in previous studies [40,51]).

An Oxipres ML 3,047,312 machine (Mikrolab Aarhus, Aarhus, Denmark) was used to determine oxidative stability. This equipment is a modification of the high-pressure oxygen bomb method (by standard ASTM D942), the principle of which is to oxidize substrates with oxygen. Enough sample was weighed into glass containers to contain 5 g of substrate. After that, each glass container was placed in an oxygen bomb, which was hermetically sealed, then degassed, and filled with oxygen up to a pressure of 5 bar. Sealed oxygen cylinders were left for 30 min to check for any pressure loss. Oxygen balls were placed in the equipment, and they began to heat and take pressure measurements (in bars). The duration of the test was 24 h, at a high pressure of 8–8.5 bar and 120 °C temperature. The consumption of oxygen during the test, as the fatty substances in the samples are oxidized, causes a pressure drop. The oxidation induction period is read at the moment when the pressure begins to decrease significantly (the point of excess of the graph curve). The protection factor (PF) is calculated according to the formula *PF* = *IPx · IPk*, where *IPx* is the induction period for the sample with antioxidant additives, hours; *IPk* is the induction period for the sample without antioxidant additives or the “blank sample”, hours. Using that formula, the oxidative stability was determined and compared among samples. 

### 3.8. In Vivo Analysis of Bioavailability

#### 3.8.1. Animals and Treatment

The experimental procedures were performed in accordance with European Community guidelines and local laws and policies. Approval for the experimental procedures was obtained from the Latvian Animal Protection Ethical Committee, Food and Veterinary Service in Riga, Latvia, Approval No. 132/2022.

For the study, a total of 21 male Wistar rats weighing 300 g were selected. These rats were then randomly divided into five groups, each consisting of four to five animals. They were housed in the animal facilities at LIOS, Latvia, under standard conditions, which included a temperature range of 21–23 °C and a 12 h light–dark cycle. The rats were provided with unlimited access to food (R70 diet, Lantmännen Lantbruk, Kimstad, Sweden) and water. Prior to the treatment, the animals were given an acclimatization period of at least one week. On the day of treatment, the rats were weighed before any administration took place. Throughout the study, no drugs other than the tested product were administered to the rats. The rats used in this study were obtained from Tartu University in Estonia.

#### 3.8.2. Administration

Various formulations of BBE were administered orally to rats at various doses (Table 6). The selected doses of each formulation were calculated to administer the following doses of BBE compounds: 76 mg/kg of betulin, 10 mg/kg of lupeol, and 2.9 mg/kg of betulinic acid; according to these values, the oleogel, oil suspension, and water suspension contain 8% BBE consisting of 52% betulin, 7% lupeol, and 2% betulinic acid, but the hydrogel contains 4% BBE consisting of 60% betulin, 8% lupeol, and 3% betulinic acid. The contents of betulin, lupeol, and betulinic acid in BBE were determined in a previous study by gas chromatography with a flame ionization detector using a Shimadzu Nexis GC 2030 apparatus [39]. The rats in Groups 1–4 were administered a full dose once, but the rats in Group 5 were administered a half dose daily for 7 days. The administration scheme can be found in the Appendix A.

#### 3.8.3. Blood Sampling

Blood samples were collected in tubes containing heparin. The blood (50–100 µL) was sampled from the tail vein at specific time intervals (2, 4, 6, 8, and 24 h) following the peroral administration of the dispersed systems. The tubes were subjected to centrifugation at 10,000× *g*-force at a temperature of +4 °C for a duration of 3 min. The resulting plasma was carefully collected and stored at −20 °C until further analysis. In addition, blood samples were collected from rats belonging to Group 5 after both the initial and final administrations.

#### 3.8.4. Analysis of Clinical Chemistry Markers in Rat Plasma Samples

Kits from the Instrumentation Laboratory were used to measure alanine aminotransferase (ALAT), blood urea (BUN), and plasma creatinine levels in accordance with the instructions specified by the manufacturer. 

#### 3.8.5. LC/MS/MS Analysis

The concentrations of betulin, betulinic acid, and lupeol in rat blood plasma were assessed using liquid chromatography–tandem mass spectrometry (LC/MS/MS) with negative electrospray ionization (ESI-) in multi-reaction monitoring (MRM) mode. The compounds were examined as p-toluenesulfonyl isocianate (PTSI) derivatives after precolumn derivatization of plasma ethylacetate extracts. Sample preparation, including extraction and derivatization, was adapted from the methods described previously [49,50]. 

The bioanalysis of betulin, betulinic acid, and lupeol was performed using a Waters Acquity UPLC H-class chromatograph connected to a Waters XEVO TQ-S tandem mass spectrometer. To determine the mass spectrometry parameters, PTSI derivatives of standard solutions containing betulin, betulinic acid, and lupeol were directly infused. For chromatographical separation, a Waters Acquity UPLC HSS T3 column (2.1 × 50 mm, 1.8 µm) was utilized, with the column temperature set to 40 °C and a flow rate of 0.4 mL/min. The separation was achieved using the linear gradient elution method, where solvent A consisted of 10 mM ammonium acetate and solvent B was acetonitrile. The gradient program was as follows: at 0 min, 5% B; at 0.5 min, 5% B; at 1.5 min, 98% B; at 3 min, 98% B; at 3.5 min, 5% B; and at 5 min, 5% B. For each analysis, an injection volume of 5 µL was used.

Experimental mass spectrometry parameters were acquired through the direct infusion of PTSI derivatives of betulin, betulinic acid, and lupeol standard solutions. The MS/MS transitions observed for the betulin derivative were 835.5→196.1 and 835.5→620.6. Similarly, for the betulinic acid derivative, the transitions were noted as 652.4→196.1 and 652.4→453.5. Lastly, the lupeol derivative exhibited a single transition of 622.4→423.5. Throughout all of the transitions, the cone voltage was maintained at 120 V, while the collision energy remained constant at 50.

#### 3.8.6. Calibration Sample and Analytical Sample Preparation

To prepare the individual stock solutions of betulin, betulinic acid, and lupeol, 2 mg of each respective compound was dissolved in 1 mL of dimethylsulfoxide (DMSO). The concentration of each stock solution was 2 mg/mL. For the working solution, 5 µL of each individual stock solution was combined, and the mixture was diluted with 85 µL of DMSO. The resulting working solution had a concentration of 0.1 mg/mL for each analyte. To prepare the calibration standards, the working solution was serially diluted with DMSO. The concentrations of the analytes in the DMSO standards were 11, 22, 55, 110, 220, 550, 1100, 2750, and 5500 ng/mL. For the plasma calibration standards, 5 µL of the DMSO standards were added to 50 µL of blank rat plasma. This resulted in plasma calibration standards with analyte concentrations of 1, 2, 5, 10, 20, 50, 100, 250, and 500 ng/mL. A “zero” calibration standard was prepared by adding 5 µL of DMSO to 50 µL of blank plasma. For the analytical rat plasma samples, 5 µL of DMSO was added to 50 µL of each sample to maintain the same composition matrix as the plasma calibration standards. Tissue homogenate samples were prepared in the same manner as the plasma samples, and quantification was performed using the plasma calibration standards.

#### 3.8.7. Extraction of Analytes from Rat Plasma and Tissue Homogenates

Tissue samples were processed using an Omni Bead Ruptor 24 homogenizer (Omni International) with a ratio of 1 : 10 (*w*/*v*) in water. The homogenized samples were then centrifuged at 10,000× *g*-force, and the resulting supernatant was collected and frozen for later analysis. To extract betulin, betulinic acid, and lupeol from rat plasma and tissue homogenates, a liquid–liquid extraction method was employed. Firstly, 500 µL of ethylacetate was added to 55 µL of a dimethylsulfoxide-spiked standard or the analytical plasma or tissue homogenate samples. The samples were mixed using a vortex mixer for 1 min at 2800 rpm. Subsequently, the samples were placed on an orbital shaker for 10 min at 250 rpm. After shaking, the samples were centrifuged at 10,000× *g*-force for 10 min to separate the ethylacetate phase from the aqueous phase. Next, 450 µL of the ethylacetate phase was carefully transferred to a V-type chromatography vial. The samples were then subjected to evaporation using a rotary evaporator Genevac EZ-2 plus in “Low BP” mode at 40 °C.

#### 3.8.8. Derivatization of Plasma and Tissue Extracts

For the evaporated samples, we added 100 µL of a 5% PTSI solution in acetonitrile to carry out the derivatization of the analytes. To ensure proper mixing, the samples were vortexed for 30 s. Subsequently, the samples were left at room temperature for 10 min for the derivatization reaction to occur. After the incubation period, another vortex-mixing step of 30 s was performed. To neutralize any excess derivatizing agent, we added 100 µL of water to the samples. A brief vortex mixing step followed to ensure homogeneity, and the samples were then used for LC/MS/MS analysis. Both hydroxyl groups of betulin were derivatized, and betulinic acid and lupeol were mono-derivatized.

### 3.9. Statistical Analysis

The statistical analysis was performed using the data analysis feature in Excel. Two statistical tests (Anova: Single Factor and Correlation) were employed to examine the connections among various variables. All statistical tests were carried out at a significance level of α = 0.05, implying that any *p*-value equal to or less than 0.05 was deemed statistically significant and would result in rejecting the null hypothesis.

## 4. Conclusions

1.The bioavailability of BBE was statistically significantly improved using an oleogel obtained through a novel method of crystallization from an ethanol solution in the oil phase compared with all other dispersion systems.2.The highest concentrations of betulin and lupeol were measured after the administration of the oleogel, reaching up to 35-fold and 80-fold higher levels than those after the administration of a suspension in water, respectively.3.The bioavailability of betulin from the hydrogel was lower, while lupeol exposure was comparable after oleogel and hydrogel administration.4.Administering half a dose of the oleogel revealed dose-dependent properties, as it led to exactly 2-fold lower BBE exposure than the full-dosage group.5.Lupeol bioavailability from the oleogel was found to be 20 times higher than betulin bioavailability. Betulinic acid was not detected in most of the samples.6.The oleogel is safe for the liver and kidney at the tested dose. Betulin and betulinic acid were not detected in rat heart, liver, kidney, or brain tissues after peroral administration of the oleogel daily for 7 days. Lupeol was found in rat heart, liver, and kidney tissues.

## Figures and Tables

**Figure 1 plants-13-00145-f001:**
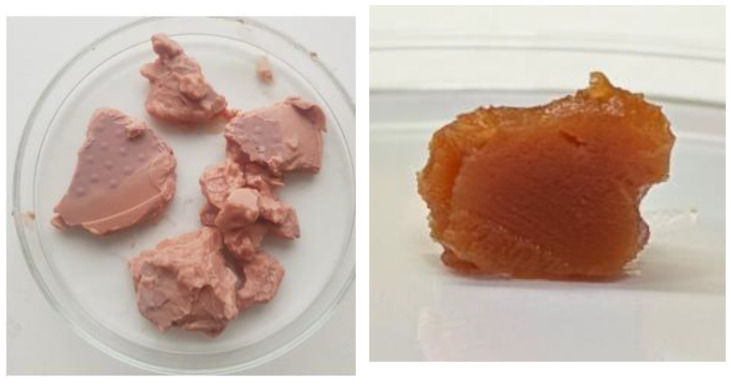
Images of BBE hydrogel (**left**) and BBE oleogel (**right**) used in this study.

**Figure 2 plants-13-00145-f002:**
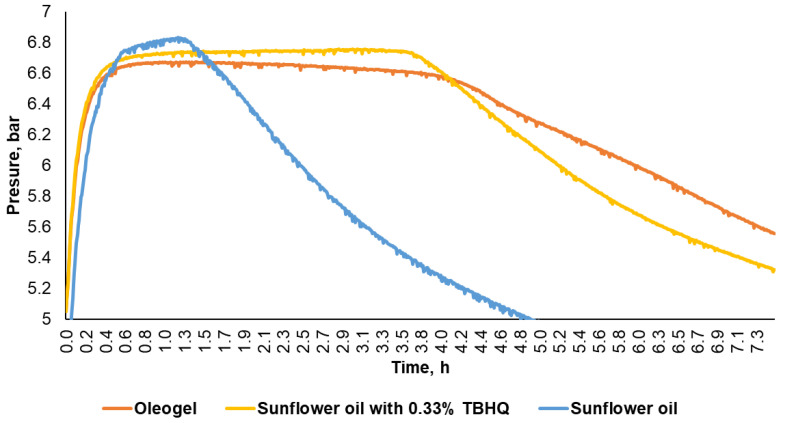
Time display of the oxidation process results of oleogel, sunflower oil, and sunflower oil with TBHQ (tertiary butylhydroquinone) additives.

**Figure 3 plants-13-00145-f003:**
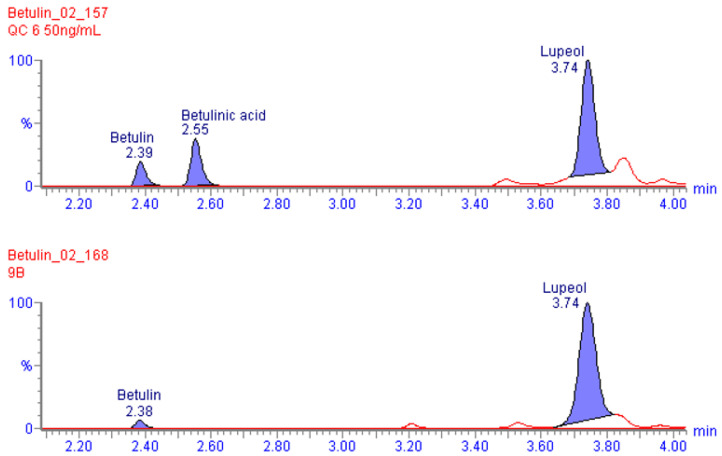
Overlay of MRM chromatograms of (**upper**) blank rat plasma sample spiked with betulin, betulinic acid, and lupeol at a concentration of 50 ng/mL and (**lower**) rat plasma sample collected 4 h after peroral administration of oleogel at a dose of 1 g/kg.

**Figure 4 plants-13-00145-f004:**
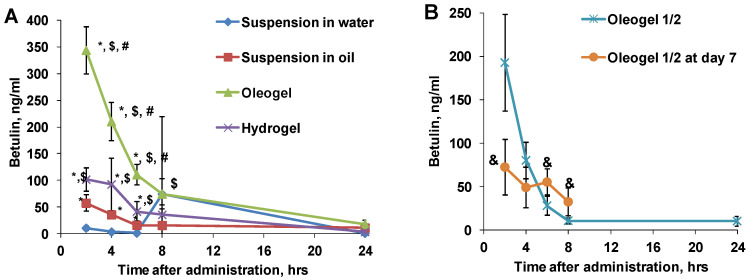
Measured concentration of betulin in rat plasma samples after peroral administration of various BBE-containing dispersed systems (**A**) and oleogel at a ½ dose after single and repeated administration (**B**). Each value was calculated as the mean ± S.E.M. of 4–5 rats. Kruskal–Wallis test * *p* < 0.05 vs. suspension in water, $ *p* < 0.05 vs. suspension in oil, # *p* < 0.05 vs. hydrogel, and & *p* < 0.05 vs. single administration of ½ dose.

**Figure 5 plants-13-00145-f005:**
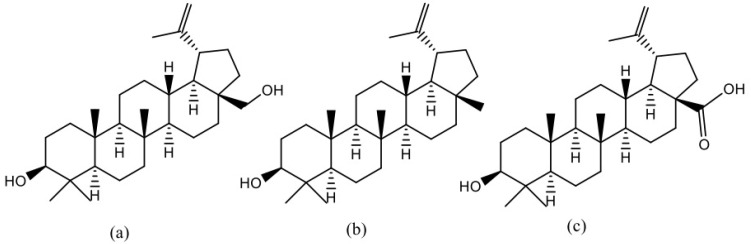
Structures of (**a**) betulin, (**b**) lupeol, (**c**) betulinic acid.

**Figure 6 plants-13-00145-f006:**
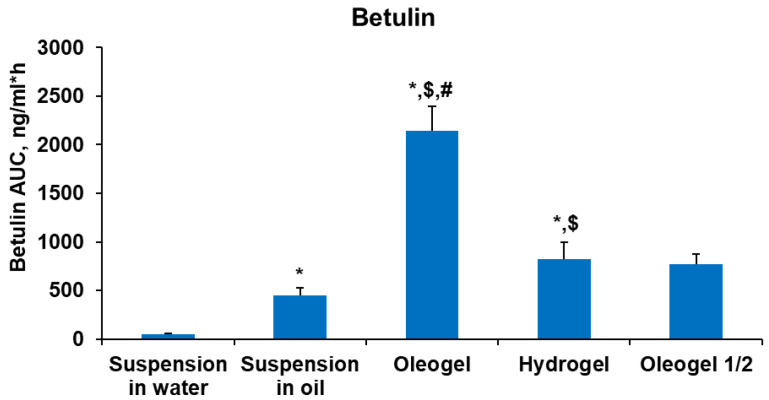
Total exposure (AUC) of betulin in rat plasma after peroral administration of various BBE-containing dispersed systems. Each value was calculated as the mean ± S.E.M. of 4–5 rats. Kruskal–Wallis test * *p* < 0.05 vs. suspension in water, $ *p* < 0.05 vs. suspension in oil, # *p* < 0.05 vs. hydrogel.

**Figure 7 plants-13-00145-f007:**
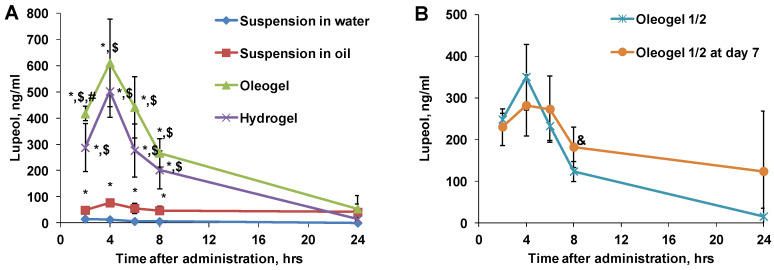
Measured concentration of lupeol in rat plasma samples after peroral administration of various BBE-containing dispersed systems (**A**) and oleogel at a ½ dose after single and repeated administration (**B**). Each value was calculated as the mean ± S.E.M. of 4-5 rats. Kruskal–Wallis test * *p* < 0.05 vs. suspension in water, $ *p* < 0.05 vs. suspension in oil, # *p* < 0.05 vs. hydrogel and & *p* < 0.05 vs. single administration of ½ dose.

**Figure 8 plants-13-00145-f008:**
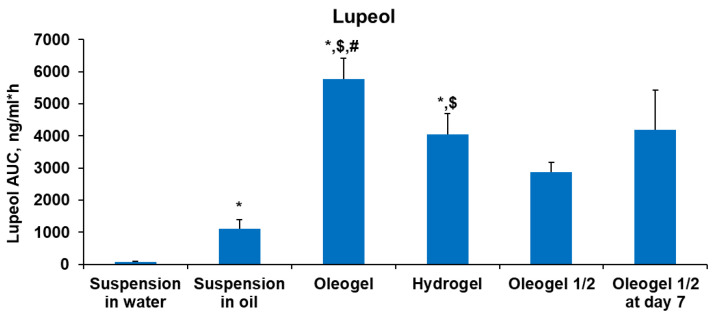
Total exposure (AUC) of lupeol in rat plasma after peroral administration of various BBE-containing dispersed systems. Each value was calculated as the mean ± S.E.M. of 4–5 rats. Kruskal–Wallis test * *p* < 0.05 vs. suspension in water, $ *p* < 0.05 vs. suspension in oil, # *p* < 0.05 vs. hydrogel.

**Figure 9 plants-13-00145-f009:**
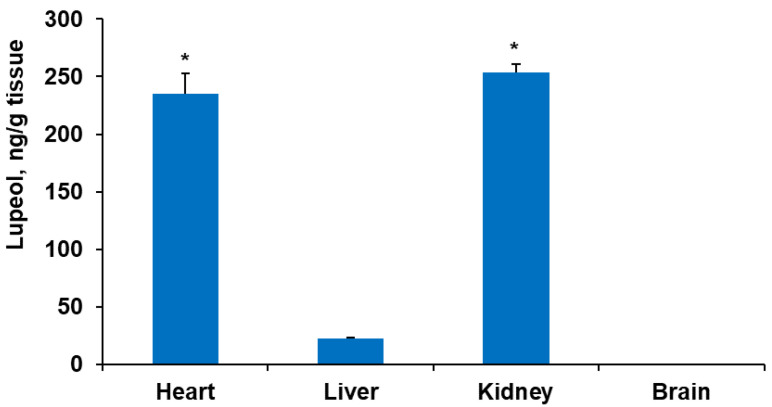
Lupeol content in rat heart, liver, and brain tissues after peroral administration of BBE oleogel half dose daily for 7 days. Kruskal–Wallis test * *p* < 0.05 vs. liver.

**Table 1 plants-13-00145-t001:** BBE particle size distribution in the different dispersion systems.

Dispersion System	BBE Powder Suspension in Water	BBE Paste-Like Hydrogel		BBE Powder Suspension in Oil	BBE Oleogel
Particle size, µm	1.210 ± 0.044	0.306 ± 0.004	D × (10)	19.8 ± 0.01	1.65 ± 0.01
			D × (50)	113 ± 3	17.0 ± 0.1
			D × (90)	205 ± 4	40.9 ± 2

**Table 2 plants-13-00145-t002:** Antioxidative properties of sunflower oil and oleogel.

Sample of Sunflower Oil	Concentration of Antioxidants, %	Induction Period, h	Protection Factor
Blank sample	-	1.3	1
Oleogel with 8% BBE	0.33 (polyphenols)	4.1	3.2
TBHQ in oil	0.33 (TBHQ)	3.5	2.7

**Table 3 plants-13-00145-t003:** Description of experimental groups.

	ALAT (Relative Units/L)	BUN (mmol/L)	Creatinine (mg/dL)
Control	33 ± 5	4.1 ± 0.5	0.15 ± 0.04
Oleogel ½ dose 7 days	27 ± 5	5.1 ± 0.5	0.08 ± 0.05

BUN—blood urea nitrogen; ALAT—alanine aminotransferase.

**Table 4 plants-13-00145-t004:** Standards and reagents.

Product	Supplier, Cat. Nr.	Lot Nr.
Betulin	Sigma Aldrich, 92648-50MG	BCCD2782
Betulinic acid	Sigma Aldrich, 91466-10MG	BCCH1050
Lupeol	Sigma Aldrich, 18692-10MG	BCCH9260
Wistar rat plasma	Innovative Research, IGWRTNaHeparin	26977
Dimethyl sulfoxide (for HPLC, ≥99.7%)	Honeywell, 34869-1L	J169AIL
Acetonitrile (≥99.9% for HPLC, gradient grade)	Honeywell, 34851-2.5L	M0910
Ammonium acetate (for LC-MS)	Sigma Aldrich, 73594-25G-F	BCCG1298
p-Toluenesulfonyl isocianate 96%	Sigma Aldrich, 189278-25G	BCCH3837
Ethanol	Kalsnava Distillery, Latvia	
THF (anhydrous, ≥99.9%)	Merck	
Pyridine (anhydrous, 99.8%)	Merck	

**Table 5 plants-13-00145-t005:** The basic composition of BBE powder.

Constituent	Amount, wt%
Betulin	52
Lupeol	7
Betulinic acid	2
Phenolic compounds	3.4
Unidentified substances	35.6

**Table 6 plants-13-00145-t006:** Description of experimental groups.

Group No.	Dispersed System	Doses, g/kg	Doses of Bioactive Compounds (Betulin; Lupeol; Betulinic Acid), mg/kg	Administrations	Animals in Group
1	Suspension in water	1.83	76.1; 10.2; 2.9	1	4
2	Suspension in oil	1.83	76.1; 10.2; 2.9	1	4
3	Oleogel	1.83	76.1; 10.2; 2.9	1	4
4	Hydrogel	3.17	76.1; 10,1; 3.8	1	4
5	Oleogel ½	0.92	38.3; 5.2; 1.5	7	5

## Data Availability

Data will be made available upon reasonable request.

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
