# Peer review of "Innovative Approach to Enhance Bioavailability of Birch Bark Extracts: Novel Method of Oleogel Development Contrasted with Other Dispersed Systems"

_plants, 2024, doi:10.3390/plants13010145_

Round 1

Reviewer 1 Report

Comments and Suggestions for Authors

Dear Author,

The manuscript entitled 

"Innovative approach to enhance bioavailability of birch bark extracts: Novel method of oleogel development contrasted with other dispersed systems" is a really interesting study. I however feel that overall presentation can be improve if you please address the following comments. 

1. Abstract need a more scientific approach rather supplying generic information.

2.     In Manuscripts, there are several grammatical and spelling mistakes author need to carefully remove mistakes. 

3.     Line 29. The word "notable" might be subjective. Consider providing evidence or a specific reference to support this claim or use a more objective term.

4.     Consider rephrasing the sentence for better clarity and structure. For instance, "Betulin, lupeol, and betulinic acid constitute the most abundant and extensively studied components of BBE [5,6]."

5.     Line 66-70. The explanation of physical hydrogels could be more detailed, particularly in relation to their impact on bioavailability.

6.     Line 91-95. The conclusion could include a brief explanation of the specific aims of this study to enhance clarity and align with the subsequent sections of the paper.

7.     Line 99. Clarify methods used to determine BBE particle sizes in water and oil phases.

8.     Line 103-106. Provide detailed differences and implications between various BBE formulations (hydrogel, BBE powder in water/oil suspension, oleogel) with specific numerical values or ranges.

9.     Line 107. Correct the spelling error ("sucsessfyl" to "successful") and elaborate on the successful preparation of hydrogel and oleogel formulations.

10.  Line 121-122: Include additional supporting evidence or references to reinforce the explanation about the relationship between smaller particle size, larger surface area of oleogel, and improved bioavailability.

11.  Line 134-135: Consider elaborating on the rationale behind the differences observed between different doses of oleogel in terms of betulin concentration.

12.  Line 179-180: The statement about the last day of administrations having a slightly lower lupeol concentration and a different PK profile lacks specific quantitative measures or statistical analysis for clarity.

13.  Line 193-196: Provide more precise quantitative data regarding the concentrations of lupeol observed in both your study and the Cháirez-Ramírez et al. study, as the current comparison lacks specific numerical values, making it challenging to evaluate the mentioned increase in bioavailability accurately.

14.  Line 220-222. Emphasize the link between the low detection of betulinic acid in rat blood plasma and the administered doses for better context and clarity.

15.  Line 251-253. Specify the clinical chemistry markers tested for liver and kidney functionality in Table 2 to enhance clarity on the parameters assessed.

16.  Line 255-256. Briefly summarize key findings from referenced studies ([6,39,52]) to strengthen claims about oleogel safety based on previous research.

17.  Line 270. Clarify the specific temperature maintained during the boiling process of the ethanol-bark mixture to ensure accurate replication of the extraction method.

18.  Line 276. Specify the temperature conditions during the drying process in the oven to ensure consistent results in obtaining the paste-like substance.

19.  Line 308-310. Specify further details about the homogenization process for preparing the oleogel to aid in reproducibility.

20.  Line 317: Please provide more details regarding the characteristics of the dried BBE and its preparation before dispersion in water to ensure reproducibility and clarity in the experimental setup

21. The similarity index is 20% please reduce that

best regards,

Comments on the Quality of English Language

 In Manuscripts, there are several grammatical and spelling mistakes author need to carefully remove mistakes. 

Author Response

Dear Reviewer, 

Thank you for your valuable suggestions and corrections to make the manuscript better!

The manuscript entitled 

"Innovative approach to enhance bioavailability of birch bark extracts: Novel method of oleogel development contrasted with other dispersed systems" is a really interesting study. I however feel that overall presentation can be improve if you please address the following comments. 

  1. Abstract need a more scientific approach rather supplying generic information.

Abstract improved

  1. In Manuscripts, there are several grammatical and spelling mistakes author need to carefully remove mistakes. 

The manuscript has been revised by an English language specialist and grammatical and typographical errors have been corrected.

  1. Line 29. The word "notable" might be subjective. Consider providing evidence or a specific reference to support this claim or use a more objective term.

Word “notable” is taken out of the sentence.

  1. Consider rephrasing the sentence for better clarity and structure. For instance, "Betulin, lupeol, and betulinic acid constitute the most abundant and extensively studied components of BBE [5,6]."

The sentence is rephrased.

  1. Line 66-70. The explanation of physical hydrogels could be more detailed, particularly in relation to their impact on bioavailability.

Updated the manuscript with some information about physical hydrogens that may affect bioavailability (line 72-80) and included an additional reference.

  1. Line 91-95. The conclusion could include a brief explanation of the specific aims of this study to enhance clarity and align with the subsequent sections of the paper.

The description of the purpose of the study was supplemented (line 103-106)

  1. Line 99. Clarify methods used to determine BBE particle sizes in water and oil phases.

The different methods for determining particle sizes in the water and oil phases are explained (line 105-106)

  1. Line 103-106. Provide detailed differences and implications between various BBE formulations (hydrogel, BBE powder in water/oil suspension, oleogel) with specific numerical values or ranges.

The numerical values are summarized in Table 1

  1. Line 107. Correct the spelling error ("sucsessfyl" to "successful") and elaborate on the successful preparation of hydrogel and oleogel formulations.

The word successful was corrected and the text was supplemented with the conditions for creating oleogel and hydrogel formulation (line 127).

  1. Line 121-122: Include additional supporting evidence or references to reinforce the explanation about the relationship between smaller particle size, larger surface area of oleogel, and improved bioavailability.

Additional references were included in the manuscript confirming the effect of particle size and surface area on bioavailability (Line 161). This information is also covered in the introduction (line 55-56)

  1. Line 134-135: Consider elaborating on the rationale behind the differences observed between different doses of oleogel in terms of betulin concentration.

Included in Line 176-177

  1. Line 179-180: The statement about the last day of administrations having a slightly lower lupeol concentration and a different PK profile lacks specific quantitative measures or statistical analysis for clarity.

Graphs 5 and 7 were divided into 2 graphs for easier perception of the PK profiles of different doses. Added statistical analysis information.

  1. Line 193-196: Provide more precise quantitative data regarding the concentrations of lupeol observed in both your study and the Cháirez-Ramírez et al. study, as the current comparison lacks specific numerical values, making it challenging to evaluate the mentioned increase in bioavailability accurately.

Supplemented with quantitative data (line 253-255)

  1. Line 220-222. Emphasize the link between the low detection of betulinic acid in rat blood plasma and the administered doses for better context and clarity.

Line 285-289

  1. Line 251-253. Specify the clinical chemistry markers tested for liver and kidney functionality in Table 2 to enhance clarity on the parameters assessed.

Information added. Below Table 2 abbreviations of markers are explained: “BUN – blood urea nitrogen, ALAT - alanine aminotransferase. Markers are mentioned in section 3.8.4.

  1. Line 255-256. Briefly summarize key findings from referenced studies ([6,39,52]) to strengthen claims about oleogel safety based on previous research.

Information updated (Line 321-323)

  1. Line 270. Clarify the specific temperature maintained during the boiling process of the ethanol-bark mixture to ensure accurate replication of the extraction method.

Information updated (line 339)

  1. Line 276. Specify the temperature conditions during the drying process in the oven to ensure consistent results in obtaining the paste-like substance.

Information updated (line 346)

  1. Line 308-310. Specify further details about the homogenization process for preparing the oleogel to aid in reproducibility.

Information updated (line 378-383)

  1. Line 317: Please provide more details regarding the characteristics of the dried BBE and its preparation before dispersion in water to ensure reproducibility and clarity in the experimental setup

Information updated (line 346-349)

  1. The similarity index is 20% please reduce that

Corrections and rewording have been made in the text to reduce the similarity

Reviewer 2 Report

Comments and Suggestions for Authors

Laura Andze et al have submitted the amnuscript entitled 'Innovative approach to enhance bioavailability of birch bark extracts: Novel method of oleogel development contrasted with other dispersed systems'. After close evaluation of the paper I recommend revision according to next points:

1.I suggests to enreach the abstract with real results. Such as dose which was not toxic, etc.

2. In Introduction: Beside anti-inflammatory, antioxidant, and anticancer properties, birch bark extract was effective in therapy for chronic hepatitis C in pilot study.

3 The bioavailability of betulin was improved by preparation of
nanosystems containing betulin for inhalation administration. This lead to the dissolution of 80% of betulin in 1 h in vitro. Please compare your results with above mentioned.

 4. Recently, pharmacokinetics and tissue disposition of nanosystem-entrapped Betulin after endotracheal administration to rats was reported. Please compare your results with published.

5. It would be logical to start section Results and discussion with the short description of oleogels and hydrogels. What was a difference? It must be clarified at the beginning of sectiom.

6. Please provide validation parameters for LC/MS/MS method for simultaneous determination of BBE

7. Please provide typical chromatograms for LC/MS/MS analysis of BBE.

8. In Sect. 2.3 please indicate administration route to animals.

9. In Fig. 1 please indicate statistical significance. The correct interpretation of data in Fig.1 is possible after statistical significance analysis.

10. In Fig. 2 please indicate statistical significance. The correct interpretation of data in Fig.1 is possible after statistical significance analysis.

11. The statistical significance is lacking in Fig 3 and 5. Please correct discussion taking in account statistical significance.

12.In Scet 3.3. how long suspension/ emulsion were stable?

13. How authors jave confirmed that oleogel was prepared? How oleogel was stabilized against oxidation. Oxidation may destroy oleogel in short time.

14. Please provide the number of Ethical Committee approval.

15. In Sect 3.2.7 please indicate administration route.

16. Please indicate a volume of blood for sampling.

17. According to  Set. 3.7.4. 'Alanine aminotransferase (ALAT), blood urea (BUN) and creatinine in plasma were
364 determined...' but no data are provided in the text.

18. The conclusion must be based on statistics and statistical significance. Please revise.

Author Response

Dear Reviewer, 

Thanks for the helpful suggestions and questions to make the manuscript better.

Laura Andze et al have submitted the amnuscript entitled 'Innovative approach to enhance bioavailability of birch bark extracts: Novel method of oleogel development contrasted with other dispersed systems'. After close evaluation of the paper I recommend revision according to next points:

1.I suggests to enreach the abstract with real results. Such as dose which was not toxic, etc.

Abstract improved

  1. In Introduction: Beside anti-inflammatory, antioxidant, and anticancer properties, birch bark extract was effective in therapy for chronic hepatitis C in pilot study.

The specific study was included in the introduction and in the references.

3 The bioavailability of betulin was improved by preparation of nanosystems containing betulin for inhalation administration. This lead to the dissolution of 80% of betulin in 1 h in vitro. Please compare your results with above mentioned.

We do not think it is fair to compare studies where betulin was administered by inhalation with our study where administration was oral, but we included this study reference to support the effect of particle size on bioavailability (line 131)

  1. Recently, pharmacokinetics and tissue disposition of nanosystem-entrapped Betulin after endotracheal administration to rats was reported. Please compare your results with published.

A comparison is included in the manuscript (Line 154-167)

  1. It would be logical to start section Results and discussion with the short description of oleogels and hydrogels. What was a difference? It must be clarified at the beginning of sectiom.

The manuscript includes visual images of the hydrogel and oleogel, as well as a brief description (section 2.1.)

  1. Please provide validation parameters for LC/MS/MS method for simultaneous determination of BBE

Validation was not performed. The limit of quantification (LOQ) was 5ng/mL for betulin and betulinic acid and 10ng/mL for lupeol. Linear detection range was 5-500ng/mL for all analytes. See paragraph 2.2.

  1. Please provide typical chromatograms for LC/MS/MS analysis of BBE.

The characteristic chromatograms are included in 2.2. in the section

  1. In Sect. 2.3 please indicate administration route to animals.

Peroral administration is already mentioned in the Abstract, Introduction, Method sections and in each figure legend.

  1. In Fig. 1 please indicate statistical significance. The correct interpretation of data in Fig.1 is possible after statistical significance analysis.

Statistical significance was added to Figure 2 (1):

  1. In Fig. 2 please indicate statistical significance. The correct interpretation of data in Fig.1 is possible after statistical significance analysis.

Statistical significance was added to Figure 3 (2)

  1. The statistical significance is lacking in Fig 3 and 5. Please correct discussion taking in account statistical significance.

Statistical significance was added to Figures 4-6 (3-5)

12.In Scet 3.3. how long suspension/ emulsion were stable?

Suspensions was not stable more than few minutes. Oleogel and hydrogel is stable for months.

  1. How authors jave confirmed that oleogel was prepared? How oleogel was stabilized against oxidation. Oxidation may destroy oleogel in short time.

We included our results of oxidative stability investigation in manuscript – section 2.2.

  1. Please provide the number of Ethical Committee approval.

The number of Ethical Committee approval is 132/2022. Information included in Institutional Review Board Statement and 3.7.1. section.

  1. In Sect 3.2.7 please indicate administration route.

The first sentence of Section 3.7.2 was modified as follows: “Various formulations of BBE were administrated orally in rats at various doses (Table 4).”

  1. Please indicate a volume of blood for sampling.

Blood (50-100ul) was sampled from the tail vein at 2, 4, 6, 8 and 24 hours after peroral (PO) administration of dispersed systems. Information included in 3.7.3. section

  1. According to  Set. 3.7.4. 'Alanine aminotransferase (ALAT), blood urea (BUN) and creatinine in plasma were
    364 determined...' but no data are provided in the text.

These data are provided in the Table 2.

  1. The conclusion must be based on statistics and statistical significance. Please revise.

Conclusion revised based on statistic analysis.

Reviewer 3 Report

Comments and Suggestions for Authors

The paper should have a specific symmetry that can be understood by any young researchers. All the sentences given in the paper should be written in simple words so that the reader can understand it. Add more references relevant to the study.

It has been observed that the other online papers on the birch genus have mentioned species as well as there are 11 or more species of birch tree, so you should also mention the specie of Birch used in the current study. After reading the rest of the papers, I figured out that you should use triterpenes in your keywords and mention triterpenes in your introduction.

In line number 150, it is said that betulin, lupeol and betulinic acid have structural similarity. Also draw their chemical structures for readership.

Write line number 51 in terms of the correct phrase.

If LC/MS is already reported, why is it mentioned in the introduction that it is done by researcher and its chemicals are also mentioned in the Materials and Methods section that these chemicals are used for LC/MS? Where are your own results for this study (if necessary) and where are the discussions, this is not mentioned in the paper?

Also add the pictures or scheme of oleogel or hydrogel you have made.

Introduction Line No. 91 to 96 In this you have told that water and oil were used then what kind of oil have you used and why are you using them.

Table 1 in line number 103 is not table 3

All headings in Materials and Methods should also be mentioned in Results and Discussion. Cite the references for technique you are using in the material and method, whether it is a pre-reported method. what content you have taken from a paper, or whether you have invented a new method yourself describe is clearly.

In Animal Study provide the infographics of Experimental Design and Pictures. Add them in Paper or Supplementary File.

Five groups of animals, only two have been characterized and discussed why the remaining three groups have not been discussed. Explain the rat groups and give reasons of 4-5 rats written under each table. Can’t understand.

It is not even clear from your paper that which activity you have checked, add the pictures of any activity you have checked, its method and discussion as well. Either discussing about BBE or betulin, lupeol and betulinic acid it is also not clear.

The solutions prepared for calibration curve and extraction were then run through which instrument to form an excel sheet and graph?

The comparison is only between suspension in oil and oleogel. Why? Must have comparison between hydrogel and oleogel. Why you used sunflower oil instead of sesame oil and olive oil?

In line 99 correct about atomic sizes or use of was and were.

In line 143 correct the first letter is capital in “Higher”.

Add the results of hydrogel.

Correct the spellings in line 107 “sucsessfyl” and so on others as well.

Some standard deviations in figures 1 & 3 are too high. Kindly justify them.

Recommendations: Major Revisions

Comments on the Quality of English Language

It seems like some forms of AI software has been used in writing the manuscript like Quilbolt, ChatGPT etc. This needs to be clarified from the authors too. 

Author Response

Dear reviewer!

Thanks for the helpful suggestions and questions to make the manuscript better!

The paper should have a specific symmetry that can be understood by any young researchers. All the sentences given in the paper should be written in simple words so that the reader can understand it. Add more references relevant to the study.

Thanks for the suggestion. The manuscript has been revised and improved to make it easier to understand. Some more references are added. The manuscript is reviewed by an English specialist

It has been observed that the other online papers on the birch genus have mentioned species as well as there are 11 or more species of birch tree, so you should also mention the specie of Birch used in the current study. After reading the rest of the papers, I figured out that you should use triterpenes in your keywords and mention triterpenes in your introduction.

In the article, we included information about the birch species Betula Pendula, at the end of the introduction and 3.1. and 3.2. in sections. Triterpenes are mentioned in the abstract, introduction and keywords.

In line number 150, it is said that betulin, lupeol and betulinic acid have structural similarity. Also draw their chemical structures for readership.

Picture included

Write line number 51 in terms of the correct phrase.

Rephrased.

If LC/MS is already reported, why is it mentioned in the introduction that it is done by researcher and its chemicals are also mentioned in the Materials and Methods section that these chemicals are used for LC/MS? Where are your own results for this study (if necessary) and where are the discussions, this is not mentioned in the paper?

In the introduction we mentioned, how betulin, betulinic acid and lupeol are measured (references included). In Materials and method, Table 3. we summarized materials, standards and reagents used for our study. In paragraph 3.7.5. we described LC/MS/MS parameters, sample preparation parameters used for current study.

Also add the pictures or scheme of oleogel or hydrogel you have made.

Picture included

Introduction Line No. 91 to 96 In this you have told that water and oil were used then what kind of oil have you used and why are you using them.

Since the solubility of BBE in oil is much higher than in water, an oil suspension was also used to evaluate bioavailability. As indicated in the methods section, sunflower oil was used, as it is the most frequently used oil.

Table 1 in line number 103 is not table 3

Corrected

All headings in Materials and Methods should also be mentioned in Results and Discussion. Cite the references for technique you are using in the material and method, whether it is a pre-reported method. what content you have taken from a paper, or whether you have invented a new method yourself describe is clearly.

Some headings is changed in Results and Discussion to be more connected with Methods. All headings from Materials and Methods have not be included in the Results and Discussion section, e.g. the results do not describe how the blood sample is taken, nor does it describe the preparation of the dispersion systems (hence the methods section).

In section 3.7.5. it is clearly described on the basis of which previous studies the method has been created (inserted references) and further explained in detail how the chromatographic measurements were made in our study. The description is provided in such a way that it is repeatable. The aim of our study was to determine the bioavailability of betulin, lupeol and betlinic acid at the same time, so the method was approved as part of the study.

In Animal Study provide the infographics of Experimental Design and Pictures. Add them in Paper or Supplementary File.

Included as Supplement File

Five groups of animals, only two have been characterized and discussed why the remaining three groups have not been discussed. Explain the rat groups and give reasons of 4-5 rats written under each table. Can’t understand.

In the Results and Discussion section starting from paragraph 3.3 all 5 groups are compared with each other. As we mention in the legends, we used 4-5 rats in each group.

It is not even clear from your paper that which activity you have checked, add the pictures of any activity you have checked, its method and discussion as well. Either discussing about BBE or betulin, lupeol and betulinic acid it is also not clear.

In the methodological part, it is clearly described that BBE containing specific amounts of betulin, lupeol and betulinic acid has been orally administered to rats. Then, after certain time inventories, blood samples were taken from the rats and blood plasma was separated, which was further analyzed by chromatography to determine the concentration of betulin, lupeol and betulinic acid in the blood plasma, thus evaluating the bioavailability.

​The solutions prepared for calibration curve and extraction were then run through which instrument to form an excel sheet and graph?

Calibration sample preparation is described in paragraph 3.7.6. In paragraph 3.7.7 we described the extraction of analytes from blood plasma un tissue homogenates. In paragraph  3.7.8 we described derivatization of analytes in blood plasma and tissue extracts and final sample preparation steps for LC/MS/MS analysis.

The comparison is only between suspension in oil and oleogel. Why? Must have comparison between hydrogel and oleogel. Why you used sunflower oil instead of sesame oil and olive oil?

Sunflower oil is the most commonly used oil in Eastern Europe. Olive oil is less common in our everyday life, but sesame oil is exclusive and is used as a seasoning.

In line 99 correct about atomic sizes or use of was and were.

Corrected

In line 143 correct the first letter is capital in “Higher”.

Corrected

Add the results of hydrogel.

Correct the spellings in line 107 “sucsessfyl” and so on others as well.

Corrected

Some standard deviations in figures 1 & 3 are too high. Kindly justify them.

Substance intake can vary significantly from individual to individual even in experimental rats, even when subjected to equal conditions. For this reason, several animals in the same group are used to determine the bioavailability trend (increasing, decreasing, higher, lower). The mean and standard deviation are then calculated.

Round 2

Reviewer 2 Report

Comments and Suggestions for Authors

Authors have adequately addressed to my comments.

Just minor revision is required:

Reference 51 is included in The Reference list but not cited in the manuscript. Please update the text of manuscript.

Author Response

Dear reviewer, 

The reference No. 51 was on line 161. We added that reference in the introduction. Now it is reference No 23 and can be found on lines 55 and 161. 

Thank you for your efforts to improve the publication.
Happy and creative New Year!

Best regards